# Spin-optoelectronic devices based on hybrid organic-inorganic trihalide perovskites

Jingying Wang[1], Chuang Zhang[1], Haoliang Liu 🄳 [1], Ryan McLaughlin[1], Yaxin Zhai[1], Shai R. Vardeny[2], Xiaojie Liu[1], Stephen McGill[3], Dmitry Semenov[3], Hangwen Guo[4], Ryuichi Tsuchikawa[1], Vikram V. Deshpande[1], Dali Sun[1,5] & Z. Valy Vardeny[1]

Recently the hybrid organic-inorganic trihalide perovskites have shown remarkable performance as active layers in photovoltaic and other optoelectronic devices. However, their spin characteristic properties have not been fully studied, although due to the relatively large spin-orbit coupling these materials may show great promise for spintronic applications. Here we demonstrate spin-polarized carrier injection into methylammonium lead bromide films from metallic ferromagnetic electrodes in two spintronic-based devices: a 'spin light emitting diode' that results in circularly polarized electroluminescence emission; and a 'vertical spin valve' that shows giant magnetoresistance. In addition, we also apply a magnetic field perpendicular to the injected spins orientation for measuring the 'Hanle effect', from which we obtain a relatively long spin lifetime for the electrically injected carriers. Our measurements initiate the field of hybrid perovskites spin-related optoelectronic applications.

[1] Department of Physics & Astronomy, University of Utah, Salt Lake City, UT 84112, USA. [2] College of Optical Sciences, University of Arizona, Tucson, AZ 85721, USA. [3] National High Magnetic Field Laboratory, Tallahassee, FL 32310, USA. [4] Department of Physics & Astronomy, Louisiana State University, Baton Rouge, LA 70803, USA. [5] Department of Physics, North Carolina State University, Raleigh, NC 27695, USA. Correspondence and requests for materials should be addressed to D.S. (email: dsun4@ncsu.edu) or to Z.V.V. (email: val@physics.utah.edu)

F inding materials for spintronics applications that simulta-
neously possess strong spin–orbit coupling (SOC) for effi-
cient spin manipulation, long-spin relaxation time for
efficient spin transport, and strong photoluminescence (PL)
emission for spin-optoelectronic applications, has been a chal-
lenge. Hybrid organic–inorganic trihalide perovskites (OITP) are
emerging semiconductors which have recently attracted intense
research interest for optoelectronic applications[1,2]. The OITP
such as MAPbX₃, where MA is methylammonium and X is a
halogen, may combine the advantages of both organic and
inorganic semiconductors for spin-optoelectronic applications.
These compounds are grown using solution-based low-tempera-
ture methods, which lead to easy fabrication and flexible device
engineering[3]. Moreover, they have synthetically tunable electronic
and optical properties with strong PL emission. Also due to the
heavy atoms in their building blocks, the OITP possess a rela-
tively large SOC. Indeed, the recently obtained Rashba-splitting in
the OITP[4,5], the optical spin selection rules[6,7], and magnetic field
effect[8] caused by strong SOC have initiated spintronics research
avenue for these compounds. So far, however no spin-related
optoelectronics device application based on OITP has been
reported. Thus, demonstration and studies of spintronic devices
based on this class of semiconductors are intriguing.

In order to realize spin-related optoelectronic devices based on
OITP, electrical spin injection from ferromagnet (FM) electrodes
into OITP films should be demonstrated. Here, we successfully
demonstrate spin injection from FM electrode into MAPbBr₃,
detected by both optical and electrical means. For the optical
detection, we have used the circularly polarized electro-
luminescence (EL) emission from MAPbBr₃ film in a spin light
emitting diode (spin-LED) device based on a FM electrode. Spin-

LED devices, first realized using III–V semiconductors[9,10], were
considered to be strong evidence for confirming spin injection
into various semiconductors[11–13]. The confirmed spin injection
ability shows that the OITP successfully overcome the con-
ductivity mismatch problem[14]. In addition, we also demonstrate
electrical spin injection into OITP using spin valve (SV) devices
based on MAPbBr₃ interlayer, where giant magneto-resistance
(GMR) of up to about 25% has been achieved. Importantly, we
also observed the "electrical Hanle effect" in the SV devices, from
which we obtained at cryogenic temperatures a spin lifetime, $\tau_s =$
936 ± 23 ps for the injected spin 1/2 holes in MAPbBr₃.

## Results

**Optical spin injection and Hanle effect in MAPbBr₃.** To
demonstrate spin injection from FM electrodes, we first studied
circularly polarized PL emission in MAPbBr₃ thin film at low
temperature, in order to show that it maintains "optical spin
alignment"[15]. We prepared polycrystalline films of MAPbBr₃ by
spin casting (see Methods). Scanning electron microscopy (SEM)
illustrated an average grain size of around 100 nm in the film,
whereas the X-ray-diffraction (XRD) pattern was similar to that
in the literature (Supplementary Fig. 1 & Supplementary Note 1)
[16]. The films were excited using circularly polarized ($\sigma^+$) cw laser
beam at 532 nm, and we measured the circular PL components,
PL($\sigma^+$) and PL($\sigma^-$) (Fig. 1a) to obtain the degree of circular
polarization, which is defined as $P_{PL} = [PL(\sigma^+) - PL(\sigma^-)]/[PL
(\sigma^+) + PL(\sigma^-)]$ at 10 K; we obtained $P_{PL}$ of 3.1% (see Fig. 1b).
The exciton binding energy in the orthorhombic phase of
MAPbBr₃ is about 30 meV[17] so that the majority of the photo-
excitations at steady state conditions at 10 K are excitons. The

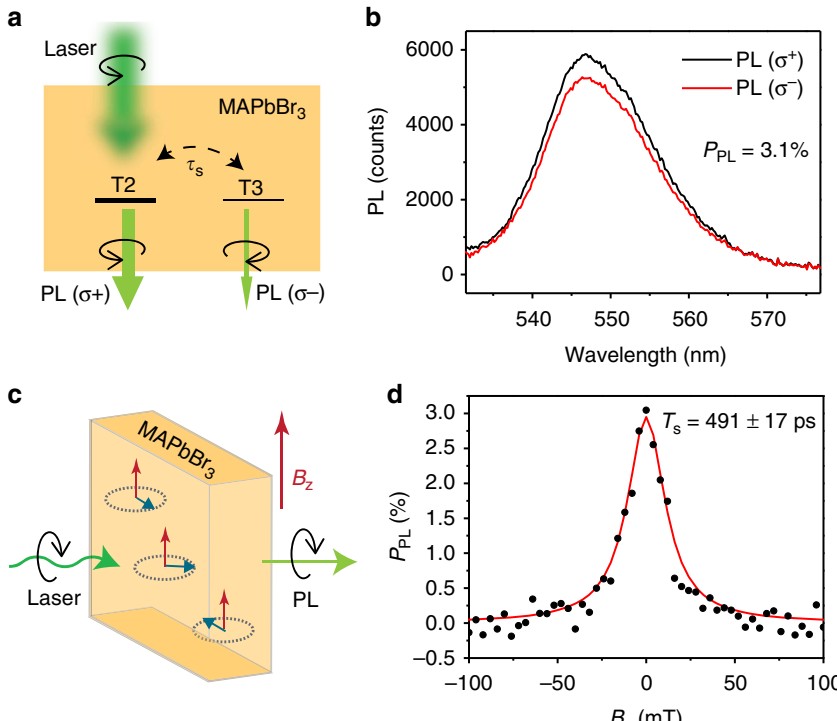

**Fig. 1** Circular polarization of the PL and optical Hanle effect in MAPbBr₃. **a** Schematic of the circularly polarized excitation ($\sigma^+$) and polarized PL emission
components, PL($\sigma^+$) and P($\sigma^-$) in MAPbBr₃. T2 and T3 are two exciton states that can emit circularly polarized PL and have spin-dependent optical
transitions near the MAPbBr₃ optical gap. Here PL($\sigma^+$) > PL($\sigma^-$), as marked by a heavier bar underneath the T2 label. Spin relaxation process couples the
two exciton states with spin lifetime,$\tau_s$. **b** PL($\sigma^+$) and PL($\sigma^-$) emission spectra in MAPbBr₃ at steady state measured at 10 K. The degree of circular
polarization, $P_{PL} = [PL(\sigma^+) - PL(\sigma^-)]/[PL(\sigma^+) + PL(\sigma^-)]$ of 3.1% is obtained. **c** Schematic set-up for measuring the optical Hanle effect. A magnetic field,
$B_z$ perpendicular to the PL light propagation direction is applied, which causes spin precession that leads to diminishing circular polarization of PL. **d** The
$P_{PL}(B_z)$ response measured at 10 K. The red solid line is a fit using Eq. (1), from which we obtained an effective exciton spin lifetime, $T_s = 491 \pm 17$ ps

photogenerated excitons may populate four closely spaced states. These are three triplets, T1–T3, where T2 and T3 emit circularly polarized PL, and a dark (D) singlet exciton[18,19]. The non-zero $P_{PL}$ validates the optical spin selection rules in MAPbBr$_3$, which allows optical orientation and detection of spin-polarized excitations, as in other direct gap semiconductors such as GaAs[20].

The polarization degree, $P_{PL}$ is determined by the two spin-polarized excitons (T2 and T3) with population $N^{+(−)}$ at steady state, where $P_{PL} = \frac{N^+ - N^-}{N^+ + N^-}$. $P_{PL}$ is related to the exciton spin lifetime, $\tau_s$ and lifetime $\tau$ by the relation: $P_{PL} = \frac{\eta}{1 + \tau/\tau_s}$[15], where $\eta$ is the initial polarization ratio after excitation. Considering the long exciton lifetime (of the order of tens ns) in MAPbBr$_3$[2,21] (Supplementary Fig. 2 & Supplementary Note 2), we conclude from the obtained $P_{PL}$ value that $\tau_s$ cannot be very short.

When an external magnetic field, $B_z$ is applied in the direction perpendicular to the spin orientation, the exciton spin precesses around $B_z$ direction, as shown in Fig. 1c; this leads to quenching of $P_{PL}$ (called optical Hanle effect). Figure 1d illustrates that $P_{PL}$ indeed decreases with $B_z$, which is traditionally described by the relation[22]:

$$P_{PL}(B_z) = \frac{P_{PL}(B_z = 0)}{1 + (\omega_L T_s)^2} \quad (1)$$

where $\omega_L = \mu_B g_{ex} B_z / \hbar$ is the Larmor frequency. We have measured the $g$-factor value, $g_{ex}$ of exciton in MAPbBr$_3$ by magnetic circular dichroism (Supplementary Fig. 3 & Supplementary Note 3) to be $g_{ex} = 1.254$. In Eq. (1), $T_s = 1/\left(\frac{1}{\tau} + \frac{1}{\tau_s}\right)$ is the exciton effective spin lifetime. The red solid line in Fig. 1d is a fit using Eq. (1), from which we obtained $T_s = 491 \pm 17$ ps in MAPbBr$_3$ at 10 K. Taking $\tau = 15$ ns[4], we calculate from the

measured $T_s$ the value $\tau_s = 508 \pm 17$ ps for the exciton spin lifetime in MAPbBr$_3$.

**Spin-LED device based on MAPbBr$_3$.** Since MAPbBr$_3$ shows circularly polarized PL emission, we may use it to probe spin injection from FM electrode by studying circularly polarized EL emission in a spin-LED device. We have therefore fabricated MAPbBr$_3$ spin-LED based on the half metal FM electrode La$_{0.63}$Sr$_{0.37}$MnO$_3$ (LSMO), which has nominally 100% spin polarization at the Fermi energy[23]. A thin MAPbBr$_3$ film was spin coated on the LSMO substrate, which serves as an anode that injects spin-polarized holes. As a cathode we used an Al film coated on top of a thin layer of the organic small molecule, 2,2′,2″-(1,3,5-benzinetriyl)-tris (1-phenyl-1-H-benzimidazole) (TPBi), which serves as the electron transport layer. The device structure is illustrated in Fig. 2a. An external magnetic field was applied perpendicular to the electrodes, which is aligned along the propagation direction of the collected EL emission. Figure 2b shows the working principle of the spin-LED device; spin-polarized holes are injected from the ferromagnetic LSMO, while unpolarized electrons are injected from the non-magnetic Al electrode. Due to the relatively large exciton binding energy of 30 meV at 10 K, the injected e–h pairs form excitons according to the optical selection rules, then emit EL light (see Fig. 2b). Because of the imbalance in the spin populations of the injected holes, EL emission shows circular polarization.

Typical electrical characteristics response of the fabricated MAPbBr$_3$ spin-LED device is shown in Fig. 2c. A clear rectifying behavior starting from a bias voltage, $V = 5$ Volt is seen. The EL–$V$ response shows a turn-on voltage at 8 Volt; bright green light can be seen from an area of 0.5 mm × 3 mm (Fig. 2d inset). At $V = 9$ Volt and 10 K, we measured a strong EL emission band at

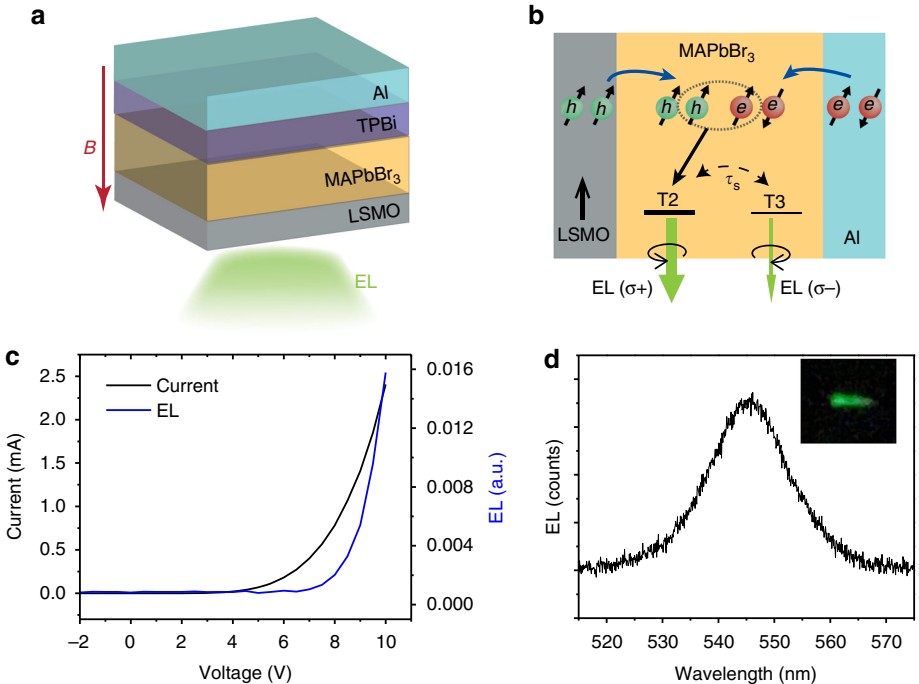

**Fig. 2** Spin-LED device based on MAPbBr$_3$. **a** Schematic of the spin-LED device structure. The "half metal" LSMO serves as FM anode that is capable of injecting spin-polarized holes into the MAPbBr$_3$ interlayer. The TPBi molecule thin film serves as the electron transport layer capped by an Al cathode. **b** Working principle of the spin-LED. Spin-polarized holes injected by the FM anode form e–h pairs with electrons injected by the nonmagnetic electrode according to the "optical spin selection rule", which subsequently form spin-polarized excitons, T2 that emit circularly polarized electroluminescence, EL ($\sigma^+$). Due to spin relaxation, T3 state may be also populated emitting EL($\sigma^-$). Here EL($\sigma^+$) > EL($\sigma^-$), as marked by a heavier bar underneath the T2 label. **c** Typical $I$–$V$ and EL–$V$ responses of the MAPbBr$_3$-based spin-LED measured at 10 K. **d** The resulting EL spectrum; the picture in the inset shows the green EL emission from the device

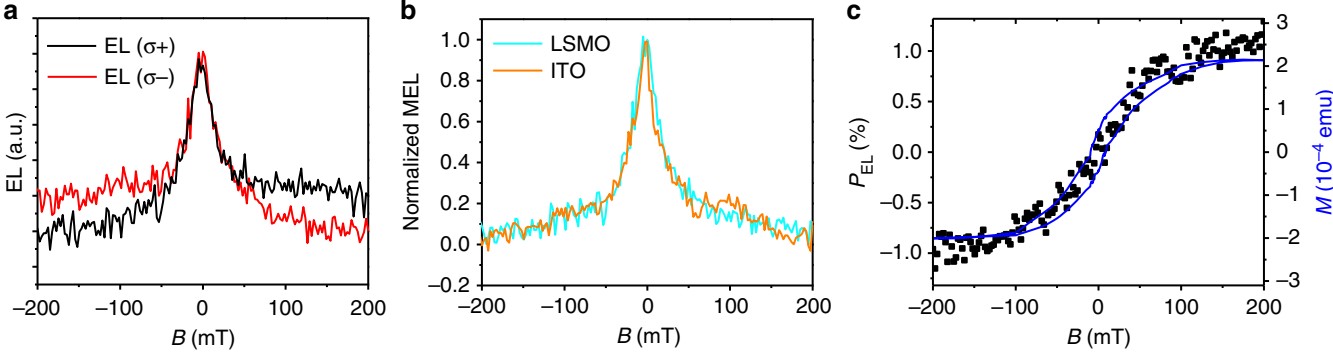

**Fig. 3** Circularly polarized electroluminescence emission from MAPbBr$_3$ spin-LED. **a** EL($\sigma^+$) and EL($\sigma^-$) circularly polarized EL emission as a function of an applied magnetic field measured at 10 K and applied bias of 9 V. **b** The sum of the right and left circular polarized EL($B$) response of the spin-LED, compared with MEL($B$) response of a traditional LED with no FM electrodes, where the LSMO was replaced by indium tin oxide (ITO). **c** The degree of circular polarization in the EL($B$) response. The blue solid line is the magnetization hysteresis loop of the LSMO electrode measured using SQUID magnetometry

548 nm (Fig. 2d), which is consistent with the PL spectrum (Fig. 1b), and with traditional LEDs based on MAPbBr$_3$[2].

The two circularly polarized EL components, EL($\sigma^+$) and EL ($\sigma^-$) were measured when the out-of-plane magnetic field aligns the LSMO magnetization, as seen in Fig. 3a. A clear difference between EL($\sigma^+$) and EL($\sigma^-$) responses versus $B$ is obtained. We note that the EL($B$) response is a superposition of two components. (i) An intrinsic, symmetric magneto-EL (MEL) response, which is the same for the EL($\sigma^+$) and EL($\sigma^-$) responses (Fig. 3b). This response is due to a spin-mixing process within the e–h pairs[8], and is the same as in an ordinary MAPbBr$_3$-based LED (with no FM electrodes) that we also fabricated (see Fig. 3b). (ii) An asymmetric MEL($B$) response which is different for the EL ($\sigma^+$) and EL($\sigma^-$) emissions. We calculated the EL degree of circular polarization, $P_{EL}$ using the relation $P_{EL} = \frac{EL(\sigma^+) - EL(\sigma^-)}{EL(\sigma^+) + EL(\sigma^-)}$; its response versus magnetic field $B$ is plotted in Fig. 3c. A maximum $P_{EL}$ value of 0.8% is obtained at $B$ larger than 150 mT. For comparison purpose, we also plot in Fig. 3c the out-of-plane magnetization $M(B)$ response of LSMO electrode. It is clear that the obtained $P_{EL}(B)$ response follows the magnetization response of LSMO. This unambiguously proves that the circular polarization of the EL is caused by spin injection from the LSMO electrode, since $P_{EL}(B)$ is proportional to the spin polarization of the injected holes from LSMO electrode, which, in turn, is related to LSMO magnetization response. A slight deviation, $\Delta P(B)$ of the $P_{EL}(B)$ response from the $M(B)$ response of the LSMO electrode is seen at large $B$. This is caused by contribution from field induced Zeeman splitting[11], and magnetic circular dichroism of the LSMO electrode. These responses were measured separately to be smaller than 0.1%, in agreement with the deviation response. (Supplementary Fig. 4 & Supplementary Note 4).

Similar to the circular polarization degree, $P_{PL}$ in MAPbBr$_3$ films discussed above, $P_{EL}$ of the spin-LED can be expressed by the relation:

$$P_{EL} = \frac{N^+ - N^-}{N^+ + N^-} = P_s \frac{\eta}{1 + \tau/\tau_s} = P_s P_{PL} \qquad (2)$$

where $P_s$ is the spin injection efficiency of the FM electrode, $\eta$ is the initial spin polarization in MAPbBr$_3$. Assuming $\eta$ is the same for PL and EL measurements, from the obtained $P_{EL}$ and $P_{PL}$ expressions we get $\frac{P_{EL}}{P_{PL}} = P_s = 26\%$ for the LSMO ferromagnetic electrode at 10 K. In fact, $\eta$ for the EL emission should be smaller than for PL emission at resonance excitation, namely $\eta_{EL} < \eta_{PL}$, since the spin aligned injected holes in the spin-LED may lose their spin alignment before forming e–h pairs. Therefore, 26% is a lower limit for the LSMO spin injection efficiency. Nevertheless,

this large value is surprising when taking into account the existence of the seemingly unsurmised conductivity mismatch between a metallic FM and a semiconductor, which mostly prevents spin injection[14]. This is especially true for the perovskite-based spin-LED in which we found that the spin injection efficiency is one order of magnitude higher than the value in GaAs-based spin-LEDs[11,24]. We speculate that the efficient spin injection into the MAPbBr$_3$ could be attributed to its large SOC, which is manifested by the obtained surface Rashba splitting[5] that suppresses the conductivity mismatch, and, in turn improves the spin injection at the FM/MAPbBr$_3$ interface[14].

**Spin valve device based on MAPbBr$_3$.** We have also measured spin injection into MAPbBr$_3$ film using electrical detection in a SV device. Figure 4a shows the SV device structure which is composed of two FM electrodes, namely a LSMO film grown on SrTiO$_3$(001) substrate and an evaporated cobalt film, with MAPbBr$_3$ film as spacer layer deposited using the same method as for the spin-LED device discussed above. An in-plane magnetic field $B$ was applied to manipulate the magnetization orientation of the two FM electrodes. Magnetization hysteresis loops of both LSMO and Co electrodes were recorded in situ, using the magneto-optics Kerr effect (MOKE) measured by a Sagnac interferometer[25]. As shown in Fig. 4b, at 10 K we measured the coercive field, $B_{c1} = 5$ mT for LSMO and $B_{c2}$ at around 75 mT for Co. Upon sweeping $B$, the relative magnetization orientation of the two FM electrodes changes from parallel (P) to antiparallel (AP) configuration, and vice-versa upon changing the field sweeping direction. This leads to change in the device resistance, $R$, with $R_P < R_{AP}$; which is called giant magnetoresistance (GMR). The obtained magnetoresistance (MR) ratio is then calculated from the relation: $MR = (R_{AP} - R_P)/R_P$. The MR($B$) response of a SV device based on MAPbBr$_3$ measured at 10 K is shown in Fig. 4c. We obtained a substantial maximum GMR, GMR$_{max}$ of 25%, indicating an effective spin injection into the MAPbBr$_3$ interlayer film, which is consistent with our spin-LED response. Possible artefacts such as tunneling magnetoresistance (TMR), tunneling anisotropic magnetoresistance (TAMR), and anisotropic magnetoresistance (AMR) have been ruled out by control experiments. (Supplementary Figs. 6 & 7, Supplementary Note 6 & 7).

In order to cross-check our results, we conducted the electrical Hanle effect measurements on the same SV. A perpendicular magnetic field, $B_z$ was applied to the SV device when the FM electrodes magnetization are in parallel or antiparallel configuration. Under this condition, the injected spin in the MAPbBr$_3$ interlayer from one FM electrode precesses with Larmor frequency: $\omega_L = g\mu_B B_z/\hbar$ while the carriers drift to the other FM electrode. This spin precession induces change in the relative

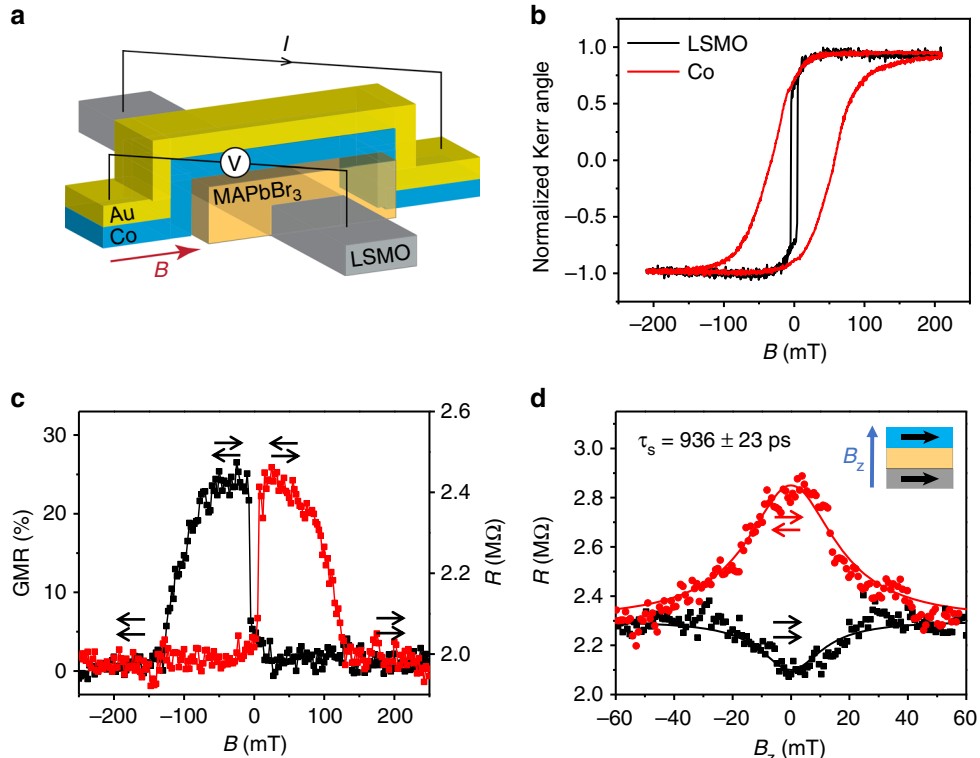

**Fig. 4** Giant magnetoresistance (GMR) and Hanle effect in MAPbBr$_3$-based spin valve. **a** Schematic of a LSMO/MAPbBr$_3$/Co spin valve device structure. **b** MOKE($B$) response of the LSMO and Co ferromagnetic electrodes as measured in-situ in the SV device at 10 K. **c** GMR($B$) response of the spin-valve measured at 10 K and applied bias voltage, $V = 0.1$ V. The obtained maximum GMR value (GMR$_{max}$) is 25%. The red and black lines represent magnetic field sweep up and down, respectively. The arrows show the mutual magnetization direction of the two FM electrodes. **d** Hanle effect of the GMR measured when a magnetic field, $B_z$ perpendicular to the FM electrodes is applied, at both parallel and antiparallel magnetization configurations. The solid lines are fits using Eq. (3), from which a spin lifetime $\tau_s = 936 \pm 23$ ps is extracted

angle between the spin polarization of injected carriers and the magnetization direction of the detecting FM electrode, thus causing change in device resistance. The precession angle, $\theta_p$ of the spin carriers that arrive at the opposing electrode is determined by the relation: $\theta_p = 2\pi t_{trans}/t_p$, where $t_p$ is spin precession period, and $t_{trans}$ is the carriers transit time (or time of flight, TOF). In disordered materials such as the OITP films, carrier transport is dispersive as demonstrated by TOF measurements[26], and this causes a broad distribution of $t_{trans}$ value. Consequently, the spin polarization of injected spin aligned carriers averages out at the detecting electrode (spin dephasing), which quenches the measured GMR of the SV device[27,28]. Figure 4d shows that GMR$_{max}$ value vanishes with increasing $B_z$ for both parallel and antiparallel SV configuration. The observed Hanle effect unambiguously demonstrates spin injection and transport in the OITP interlayer; consequently our measurements show in fact GMR type response. As a control experiment, after the Hanle measurements were performed, the in-plane MR-loop was repeated to confirm that the magnetization state of the electrodes was not tilted or flipped by perpendicular field.

To extract additional information about the spin lifetime in the MAPbBr$_3$ interlayer the electrical Hanle response, $R(B_z)$ may be fit using the 1-D spin drift-diffusion model[28]:

$$R(B_z) \propto \int_0^\infty \frac{1}{\sqrt{4\pi Dt}} \exp\left(-\frac{d^2}{4Dt}\right) \cos(\omega_L t) \exp\left(-\frac{t}{\tau_s}\right) dt \quad (3)$$

In Eq. (3), $D$ is the spin diffusion coefficient, $d$ is the MAPbBr$_3$ film thickness ($d = 207$ nm here). $\tau_s$ and $\omega_L = \mu_B g B_z/\hbar$ are the

spin lifetime and Larmor frequency of the spin 1/2 injected carriers. Since holes are the injected carriers in our SV device type, we use in Eq. (3) the $g$-factor for holes: $g_h = 0.33$[6]. Using Eq. (3) to fit the Hanle response, we obtain $\tau_s = 936 \pm 23$ ps and $D = 0.21 \pm 0.08$ cm$^2$ s$^{-1}$ for the holes in MAPbBr$_3$ at 10 K. This large spin lifetime contributes to the long spin diffusion length of around 220 nm in MAPbBr$_3$ (Supplementary Fig. 8 & Supplementary Note 8). It is interesting that the obtained $\tau_s$ for holes is larger than that for excitons.

## Discussion

In this work, we presented the study of spintronic devices based on OITP polycrystalline films. Successful spin injection from FM metallic electrodes into OITP has been demonstrated. The spin lifetime for both excitons and holes in MAPbBr$_3$ is surprisingly long despite the large SOC; this indicates that the OITP semiconductor family could be promising candidates for spintronics applications.

## Methods

**Sample preparation**. The ferromagnetic LSMO thin films were grown on SrTiO$_3$ (001) substrates by pulsed laser deposition (PLD) and patterned by wet-etch optical lithography. The LSMO electrodes were cleaned and re-used multiple times. The MAPbBr$_3$ overlayer was deposited by the spin coating method inside a N$_2$ filled glove box (O$_2$/H$_2$O < 1 ppm). The MAPbBr$_3$ precursor solution was obtained by mixing PbBr$_2$ and MABr (molar ratio = 1:1.1) in dimethyl sulfoxide (DMSO) at a concentration of 0.5 M. The perovskites solutions were stirred overnight at 50 °C before use. To prepare the perovskite thin film, the solutions were spin-coated on the O$_2$ plasma treated substrates at 4000 rpm. To achieve pinhole-free microscopic structure for the MAPbBr$_3$ film, chloroform solvent was drop-casted onto the perovskite film for a nanocrystal pinning process during spin coating (as described in ref. [2]). Finally, the perovskite films were annealed on a hot plate at 100 °C for 30 min.

**Device fabrication**. For the various devices, MAPbBr$_3$ thin film was spin coated onto the bottom LSMO electrode. After cooling down to ambient temperature, the samples were transferred back to the vacuum chamber for e-beam evaporation of other layers. For the spin-LED device, the bottom LSMO electrode was patterned as 3 × 5 mm strips. Following the MAPbBr$_3$ film deposition, a 10 nm TPBi film was deposited as the electron transport layer. Subsequently, a 100 nm aluminum film was coated as top electrode in a crossbar configuration. The typical device area was 0.5 × 3 mm. For the SV device, the bottom LSMO electrode was patterned as 0.2 × 5 mm strips. 15 nm cobalt film followed by 30 nm gold were capped as the second ferromagnetic electrode in a crossbar configuration, using a shadow mask. The typical device area in this case was 200 × 200 μm.

**Device characterization**. The out-of-plane magnetization curve of LSMO electrode in spin-LED device is measured by SQUID magnetometer at 10 K. On the other hand, magnetic hysteresis loop of two ferromagnetic electrodes is measured by a home-built Sagnac interferometer with a static DC Kerr rotation sensitivity of 20 nrad, and spot size of about 1 μm. The MOKE responses of both LSMO and Co electrodes were measured at 10 K in a perovskite-based SV configuration. This unique technique has enabled us to measure the magnetization properties of the spin-polarized interface in situ rather than bulk Co and LSMO electrodes outside the SV device. The EL and PL were measured by a silicon detector. The transport measurements were performed in a closed-cycle refrigerator with temperature ranging from 10 to 300 K by a standard four points method with Keithley 236 power supply and Keithley 2000 multimeter with an in-plane magnetic field B.

## Data availability

The data that support the findings of this study are available from the corresponding author on reasonable request.

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

## Acknowledgements

The work was supported by the Department of Energy Office of Science, Grant DE-SC0014579. The film growth facility was supported by the Center for Hybrid Organic-Inorganic Semiconductors for Energy (CHOISE), an Energy Frontier Research Center funded by the Office of Basic Energy Sciences, Office of Science within the US Department of Energy program at the University of Utah. H.G. was supported by the US Department of Energy (DOE) under Grant No. DOE DE-SC0002136. The authors thank R.A. Davidson for the SQUID measurements. The magnetic circular dichroism experiment was performed at the National High Magnetic Field Laboratory in Tallahassee, FL, which is supported by National Science Foundation Cooperative Agreement No. DMR-1157490 and the State of Florida. V.V.D. acknowledges the use of ACS PRF grant #58164 DNI10. D.Sun is grateful for support from the startup provided by the North Carolina State University.

## Author contributions

The project was planned by D. Sun and Z.V.V. The samples were prepared by J.W., X.L., and C.Z. The PL, EL, and SV measurements were conducted by J.W. The MOKE measurements were performed by R.M. The SQUID data was measured by D. Sun. The magnetic circular dichroism was measured by H.L., S.M., and D. Semenov. H.G. prepared the LSMO substrates. S.R.V. was responsible for nonlinear optics measurement. R.T. and V.V.D. were responsible for wire bonding. The picosecond measurement was done by Y. Z. The data analysis was performed by J.W. and Z.V.V. The manuscript was prepared by J.W. and Z.V.V. and discussed with all other co-authors.

## Additional information

**Competing interests:** The authors declare no competing interests.

