## [Peer Review File · Nature Communications]

Reviewers' comments:

Reviewer #1 (Remarks to the Author):

In the present submission, the authors have made effort to solve the issues we have raised before. However, the major concerns remain that 1) the board interest is still not high enough to be in a general high-impact journal like nature communication, and 2) the physics behind the mechanisms in the paper are still difficult to follow which has also been pointed out by other reviewers. For example, the controlling of the imbalance in the spin populations of the injected holes is still fishy to me, and the correlation between properties and performance and the Rashba effect still lack of sufficient experimental evidence. Although the authors claimed that the MAPbBr₃ surface has less inversion symmetry, no structural characterization on this specific sample proves that. All results are acquired from the observed device feature yet one should consider the structural complexity of the perovskite surface. Therefore the conclusions would not be so convincing. Based on the above concerns, I do not think it reaches the level to be published in Nature Commun.; however, after further clarification on the physics model, it could become suitable in more specific journals.

Reviewer #2 (Remarks to the Author):

Compared to the previous version of the manuscript, the authors have improved their measurements of the spin lifetimes, and streamlined their discussion of it. The improved manuscript is technically sound, and I find the claims of long-lived spin injection to be convincing.

One remaining weakness of this paper is the lack of a theory for the long-lived and efficient spin injection in these halide perovskites. A full answer might draw upon the presence of the Rashba effect, and, as the authors note, could easily lead to discussions outside the scope of the paper.

As it stands, the experimental findings of this paper raise questions within the field of halide perovskites, and are likely to lead to better theories on the fundamental spin and charge transport mechanisms in these materials, if the reason for the efficient spin injection can be understood. This has implications not only for spintronics of halide perovskites, which the authors are emphasizing, but perhaps equally importantly, for fully understanding their photovoltaic properties.

Detailed Response to Reviewer #1:

In the present submission, the authors have made effort to solve the issues we have raised before. However, the major concerns remain that 1) the board interest is still not high enough to be in a general high-impact journal like nature communication.

Author's response to 1): As we emphasize in the revised introduction, the spintronic properties of the hybrid perovskites have attracted substantial interest. The idea of spintronic devices based on the hybrid perovskites has been proposed by *theoretical* studies. (see for example Ref. [6]). However our work is the first *experimental* study of spintronic devices based on the hybrid perovskites family. In our work we demonstrate that our spintronics devices show great spin injection efficiency, long spin lifetime and spin diffusion length. These are some of the main ingredients that justify spintronics applications. Our work may attract more attentions and induce discussion in this field. We therefore believe that our work merits the broad attention provided by publication in *Nature Communications*.

And 2) the physics behind the mechanisms in the paper are still difficult to follow which has also been pointed out by other reviewers. For example, the controlling of the imbalance in the spin populations of the injected holes is still fishy to me.

Author's response to 2): The spin imbalance of the injected carriers is controlled by the relative density of the majority and minority carriers of the FM electrode, of which response upon the application of an external field is determined by the relative magnetization response, $M(B)$. In particular, $M \propto P_{FM} = \frac{N^\uparrow - N^\downarrow}{N^\uparrow + N^\downarrow}$, where $N^{\uparrow(\downarrow)}$ are the population of spin up (down) carriers. The spin-polarized carrier injected from ferromagnetic electrode into semiconductor, i.e. the spin polarization in the non-magnetic semiconductor is related to: $P_{SC} = P_s \cdot P_{FM}$, where P_s is the spin injection efficiency of the FM electrode. (See Ref [1] and Ref [16] for more details.) Therefore, the spin imbalance, or the spin polarization (as we call in the manuscript) follows the magnetization of ferromagnetic electrode, $M(B)$ as shown in Fig. 3c.

This spin polarization control is the fundamental property of spintronic devices. Moreover, in spin-LED devices, the polarized EL emission, which follows the track of magnetization response of FM electrode, has also been regarded as evidence of spin injection. (see Ref [11] and Ref [15] for details.) In order to clarify this point, we added some explanation of this point in the revised text.

And 3) the correlation between properties and performance and the Rashba effect still lack of sufficient experimental evidence. Although the authors claimed that the MAPbBr₃ surface has less inversion symmetry, no structural characterization on this specific sample proves that. All results are acquired from the observed device feature yet one should consider the structural complexity of the perovskite surface. Therefore the conclusions would not be so convincing.

Author's response to 3): We measured high spin injection efficiency in the fabricated spintronic devices based on MAPbBr₃, and we also demonstrated lack of inversion symmetry

and inverse Rashba-Edelstein effect at the surface of MAPbBr₃ in the last response letter. We have added these important pieces of evidence to the Supplemental Information. Therefore, we speculate that the high spin injection efficiency has some correlation of this extraordinary surface. *This is an outlook of this manuscript.* However it is not the main conclusion of our manuscript. We hope that our future study may provide sufficient incentives to follow it with more direct evidence of this point. In any case this is also an important reason the our manuscript could attract research interest from other researchers, who may further study the correlation between Rashba effect and spin injection using both theoretical and experimental tools. But at the present stage of this manuscript, as Reviewer #2 said in his/her comments, “A full answer might draw upon the presence of the Rashba effect, and, as the authors note, could easily lead to discussions outside the scope of the paper.”

As for structural characterization of the MAPbBr₃ surface, we have included SEM and XRD of the sample in the Supplemental Information. Also the MOKE measurements shown in Fig 4b were done in-situ of the device representing some magnetic properties of the MAPbBr₃/ferromagnetic electrode interface. Since the interface of metal/MAPbBr₃ is buried inside the device, most structural characterization methods are technically impossible.

Based on the above concerns, I do not think it reaches the level to be published in Nature Commun.; however, after further clarification on the physics model, it could become suitable in more specific journals.

Author’s response: We have addressed the reviewer’s comments and questions, and made changes in the revised text adding more information in the manuscript and supplemental information. We hope that the manuscript is now ready for a swift publication in *Nature Commun.*

Detailed Response to Reviewer #2:

Compared to the previous version of the manuscript, the authors have improved their measurements of the spin lifetimes, and streamlined their discussion of it. The improved manuscript is technically sound, and I find the claims of long-lived spin injection to be convincing.

One remaining weakness of this paper is the lack of a theory for the long-lived and efficient spin injection in these halide perovskites. A full answer might draw upon the presence of the Rashba effect, and, as the authors note, could easily lead to discussions outside the scope of the paper.

As it stands, the experimental findings of this paper raise questions within the field of halide perovskites, and are likely to lead to better theories on the fundamental spin and charge transport mechanisms in these materials, if the reason for the efficient spin injection can be understood. This has implications not only for spintronics of halide perovskites, which the

authors are emphasizing, but perhaps equally importantly, for fully understanding their photovoltaic properties.

Author's response: We thank the reviewer for these positive comments. His/her previous comments helped us to improve our understanding of spin lifetime of exciton and free carriers, as well as enhancing the quality of our manuscript.

REVIEWERS' COMMENTS:

Reviewer #1 (Remarks to the Author):

I am happy with the response to the comments and the revision made accordingly.

REVIEWERS' COMMENTS:

Reviewer #1 (Remarks to the Author):

I am happy with the response to the comments and the revision made accordingly.

Authors' response: We thank the reviewer for his/her comments. His/her previous comments helped us improve the quality of our manuscript.